# Automotive Cybersecurity: A Survey on Frameworks, Standards, and Testing and Monitoring Technologies

**DOI:** 10.3390/s24186139

**Published:** 2024-09-23

**Authors:** Claudiu Vasile Kifor, Aurelian Popescu

**Affiliations:** Faculty of Engineering, Lucian Blaga University of Sibiu, 55024 Sibiu, Romania; aurelian.popescu@ulbsibiu.ro

**Keywords:** automotive, vehicle, cybersecurity frameworks, cybersecurity standards, cybersecurity monitoring, cybersecurity testing

## Abstract

Modern vehicles are increasingly interconnected through various communication channels, which requires secure access for authorized users, the protection of driver assistance and autonomous driving system data, and the assurance of data integrity against misuse or manipulation. While these advancements offer numerous benefits, recent years have exposed many intrusion incidents, revealing vulnerabilities and weaknesses in current systems. To sustain and enhance the performance, quality, and reliability of vehicle systems, software engineers face significant challenges, including in diverse communication channels, software integration, complex testing, compatibility, core reusability, safety and reliability assurance, data privacy, and software security. Addressing cybersecurity risks presents a substantial challenge in finding practical solutions to these issues. This study aims to analyze the current state of research regarding automotive cybersecurity, with a particular focus on four main themes: frameworks and technologies, standards and regulations, monitoring and vulnerability management, and testing and validation. This paper highlights key findings, identifies existing research gaps, and proposes directions for future research that will be useful for both researchers and practitioners.

## 1. Introduction

Over the last 40 years, cars have undergone significant transformations to meet quality, environmental, and safety requirements, as well as to satisfy customer expectations for performance, comfort, and driver assistance. In the 1980s, core electrical systems were introduced alongside anti-lock braking systems (ABSs) and airbags. Between the 1990s and early 2000s, vehicles saw a rise in electronic control units (ECUs) within their electrical architecture, driven by a focus on production efficiency and maintenance, as well as a desire for car customization through functional features. These features could later be updated by replacing ECUs or certain sensors and actuators. As time progressed, the electrical and electronic (E/E) architecture became decentralized. Safety and comfort functions were separated, and vehicle functions were distributed among many interconnected ECUs. Each ECU was capable of processing its own data and communicating with others to implement advanced functionalities [1].

In the late 2000s and early 2010s, connected and autonomous vehicle functions were introduced, transforming vehicle systems from isolated entities into open systems capable of exchanging information with the environment, drivers, and other traffic participants. This shift led to a significant increase in the complexity of the E/E architecture with each electrical subsystem, such as braking, steering, propulsion, infotainment, and connectivity systems, incorporating between two and ten ECUs. These subsystems also began to utilize various communication protocols, including LIN, CAN, FlexRay, and MOST [2]. By the 2010s, advanced driver assistance systems (ADASs) had entered the market, featuring technologies such as emergency braking systems, lane-keeping assist systems, park assist systems, predictive forward collision warning systems, and autopilot functionality.

Today, vehicles are becoming increasingly connected through diverse communication channels, necessitating secure access for authorized users, the protection of driver assistance and autonomous driving data, and ensuring data integrity to guard against misuse or manipulation. Despite the advantages offered by these new technologies, there have been numerous intrusion incidents reported in recent years, highlighting the vulnerabilities in current systems [3,4,5,6].

Original Equipment Manufacturers (OEMs), international standard organizations, and customers face significant challenges in reducing cybersecurity risks throughout the development, production, and operation of vehicles. To maintain and enhance the performance, quality, and reliability of vehicle systems, software engineers must overcome major hurdles, including in communication diversity, software integration, testing complexity, compatibility and core reusability, safety and reliability assurance, data privacy, and software security [7]. Mitigating cybersecurity risks across these domains is a difficult challenge that requires practical solutions.

To address these challenges, establishing specific norms to standardize vehicle development, validation, and manufacturing processes has become a critical step. In 2016, the Society of Automotive Engineers (SAE) published SAE J3061, a guideline for developing secure automotive systems [8]. One of the key principles of this guideline is that cybersecurity must be integrated into the design of features from the outset, rather than being appended at the end of the development process. In 2021, the World Forum for Harmonization of Vehicle Regulations, a working group within the Sustainable Transport Division of the United Nations Economic Commission for Europe, published UN Regulation No. 155 [9]. According to this regulation, each OEM must establish and maintain a Cyber Security Management System (CSMS) to address organizational processes, responsibilities, and governance related to cybersecurity. The goal is to protect vehicles from cyber threats and attacks. OEMs are required to identify risks associated with vehicle technologies and implement measures to safeguard against them. These risk management processes must be demonstrated when OEMs seek vehicle type approval from validation authorities.

The engineering requirements for managing cybersecurity risks across the concept, product development, production, operation, maintenance, and decommissioning phases of E/E systems in road vehicles are detailed in the ISO/SAE 21434:2021 standard [10]. Starting in July 2024, this standard will apply to all newly manufactured vehicles. A key challenge for OEMs is integrating the new CSMS into the traditional automotive software development lifecycle. In the initial phase of implementing the standard, the CSMS will likely be added as an “add-on” to existing tools. Over time, automotive companies will develop solutions to more seamlessly integrate it into their development processes, similar to how safety requirements under ISO 26262 were eventually integrated [11].

A generic Safety Management System for the automotive industry is introduced in [12], where it is compared with similar systems from aviation, marine, and rail industries. Current safety standards primarily focus on systems controlled by humans. However, to support the development of autonomous vehicles, safety processes need to be updated. To address this, the ISO 21448:2022 standard (Road vehicles—Safety of the intended functionality) [13] was created.

The introduction of these new cybersecurity norms has spurred researchers to critique and suggest improvements to certain requirements [14,15]. Beyond the commercial interest in these norms, researchers are keen to provide solutions and tools to aid their implementation and contribute to the evolution of these regulations over time.

This study aims to analyze the current state of research regarding cybersecurity in general, with a particular focus on four main themes that were revealed by a VOSViewer cluster analysis: frameworks and technologies, standards and regulations, monitoring and vulnerability management, and testing and validation. It also seeks to propose development directions that will be useful for both researchers and practitioners.

## 2. Automotive Cybersecurity—A Bibliometric Analysis

A comprehensive literature search was conducted using the Preferred Reporting Items for Systematic Reviews and Meta-Analyses (PRISMA) guidelines to identify existing studies and approaches related to cybersecurity in the automotive industry (Figure 1).

The searches were conducted in December 2023 in the Clarivate Web of Science database using the following keyword strings: “cybersecurity” + “vehicle” or “cybersecurity” + “automotive”.

From the search results, we identified 801 studies, as follows: 405 proceedings papers, 337 articles, 48 review articles, and 11 documents in other categories (book chapters).

The first article on automotive cybersecurity was published in 2012. However, there has been a significant increase in the number of articles published in recent years. Approximately two-thirds of the total articles have been published within the last three years (Figure 2), reflecting the growing importance of cybersecurity in the automotive sector.

The overwhelming majority of the articles were published in English (793 articles). Only a few articles were available in other languages, including: Spanish, German, and Portuguese.

For the detailed analysis, we focused exclusively on publications classified as Articles and Reviews. To ensure that our analysis concentrated on impactful studies, we introduced an additional filter: articles must have at least two citations, which serves as a minimum level of visibility. This criterion excluded other 77 documents.

After applying this filter, we conducted a title and abstract screening, to further narrow down the studies, removing an additional 175 studies that were deemed unrelated to our research focus. This includes studies primarily centered on areas such as drones, flying vehicles, and cybersecurity for electric grid management systems used in charging electric vehicles.

The 133 articles selected for detailed review were processed using VOSViewer software, version 1.6.20 [16]. This software allowed us to perform a comprehensive analysis of the keywords and themes present in the selected studies, revealing four distinct clusters (Figure 3): Frameworks and Technologies (green, partially overlapped with yellow), Standards and Regulations (purple), Monitoring and Vulnerability Management (red), and Testing and Validation (blue)

These clusters provided a structured foundation for our detailed literature review in the following sections.

## 3. Literature Review

### 3.1. Frameworks and Technologies for Cybersecurity

#### 3.1.1. CS Frameworks

To tackle cybersecurity challenges, OEMs must adapt their governance, culture, processes, and technologies. Several articles suggest the need for a Systems of Systems (SoS) or framework approach to effectively implement CSMS and detect and mitigate threats [17,18]. A SoS approach should encompass not only the vehicle and its components, but also the cloud where vehicle data are stored, interactions with other vehicles (such as in platooning functions), road infrastructure (Vehicle-to-Road communication systems), and charging stations for electric vehicles.

Z. El-Rewini proposes a three-layer cybersecurity framework—encompassing sensing, communication, and control layers—to provide a comprehensive understanding of security threats [19]. Among these, the communication layer is identified as the most vulnerable, with the potential for significant damage.

In another study, V.K. Kukkala et al. [20] present an architecture related to Connected and Autonomous Vehicles (CAVs) and identify various vulnerabilities that could affect these systems, suggesting also possible solutions. Malware detection techniques are categorized here into signature-based, behavior-based, cloud-based, heuristic, and machine learning (ML) approaches. The authors recommend that future research on combating malware attacks should focus on lightweight cryptographic authentication, firewall systems, deep learning (DL) using offloading computation mechanisms, and software-defined security [21].

A. Khalid et al. [22] introduced the Framework for Analysis, Comparison, and Test of Standards (FACTS), which comprises four key steps: analyzing stakeholders (including government entities, battery manufacturers, battery management system (BMS) manufacturers, and OEMs), assessing their technical requirements through Threat Analysis and Risk Assessment (TARA), comparing various standards, and validating these standards using testing methods such as fuzz testing and penetration testing.

In a similar vein, S. Khalid Khan et al. [23] propose a conceptual System Dynamics (SDs) model for evaluating the cybersecurity of CAVs. This model integrates various elements such as the CAV communication framework, secure physical access, human factors, penetration levels, regulatory laws, policy frameworks, and trust across the CAV industry and the public. A Casual Loop Diagram, a method derived from system theory, is used to model the structure of SD, and to clarify the relationships between its variables.

Aldhyani et al. [24] developed a high-performance system using a ML approach to protect vehicle networks from cyber threats. Their proposed security solution was validated using a real vehicle network dataset that includes attack types like flooding, spoofing, replay attacks, and benign packets. This model can be integrated into some ECUs as software packets or, more realistically, as a dedicated Intrusion Detection System (IDS).

Wang et al. [17] designed a framework to analyze the performance of CAV platoons in various driving environments. This model aids in developing new cybersecurity requirements for emerging ECUs and crafting test scenarios for system validation.

Chandwani et al. [25] identify current security challenges and recommend solutions for mitigating cyber threats to electric vehicles and their onboard charging systems. As a use case, 6.6 kW onboard charger topologies are presented along with their potential threat vectors and corresponding countermeasures. These include: security measures for the CAN protocol (over MAC), FPGA (Field-Programmable Gate Array)-based protection, and hardware-based defense mechanisms such as short-circuit protection, digital signal processing for digital filters, and intelligent data processing algorithms for sensor signals. Additionally, Hafeez et al. [26] offer a practical solution to spoofing attacks within In-Vehicle Networks (IVNs) through ECU Fingerprinting using Parametric Signal Modeling.

Sabaliauskaite et al. [27] examined the interdependencies between safety and cybersecurity, exploring how safety measures impact cybersecurity and vice versa. They developed the TOMSAC methodology for managing trade-offs between automotive safety and cybersecurity. This includes cases of conditional dependency (where a safety requirement is a prerequisite for security), antagonism (where safety and security requirements conflict), and reinforcement (where enhancing safety also improves security or vice versa).

#### 3.1.2. CS Technologies

##### Model-Based Engineering (MBE)

Model-based engineering is widely used in the automotive industry because of the low costs of SW production, code reusability and the ability to work in complex projects [28]. It is important for model-based development tools to be checked so that they do not introduce cybersecurity vulnerabilities during software creation [10].

MBE also improves testing capabilities and testcases generation and can be easily automated if the SW development is based on MBE. Using model-based testing methods for automotive security testing could help to discover issues earlier than other manual methods (penetration test, for instance) [29,30,31]. The output of threasts assessment (attack trees) is used to create test cases that are generated automatically. An expanded model-based methodology to test SW updates over-the-air was analyzed by Kirk et al. and effective testcases were derived from attack tree results from security treat analysis [32,33].

MBE and component-based SW engineering (CBSE) can be combined as a cost-effective option to deal with the complexity of the SW. Previous studies suggest to group the subsystem (safety/non-safety systems) in security islands, separated by gateways, integrate cybersecurity HW accelerators in the new processors to sustain message encryption in real-time [28].

##### Blockchain

Previous studies explored ML and blockchain technologies as cybersecurity defense mechanisms for IVNs [34,35,36]. ML-based solutions are analyzed based on intrusion/malware detection, topology (centralized or distributed), and technical dimension (traditional ML and DL), while blockchain-based solutions are analyzed based on secure data storage, secure onboard communication, secure data access, consensual protocol (proof of work; proof of stake), and deployment/miner [37,38].

In general, blockchain-based solutions have four applications: secure data access, secure data storage, secure data transmission, and data contribution [38,39,40]. For data contribution application, it is important to mention CreditCoin, a protocol that was created in order to encourage vehicles to share information in smart vehicle networks [41].

Kim et al. [42] proposed the adoption of current blockchain technology in BMSs (battery management systems), which can be used as a cybersecurity reference for the development of battery systems. Other studies investigate how new blockchain algorithms could solve current automotive challenges, such as growing ledgers, increasing scalability, and reducing complexity and latency [37,38,43].

Blockchain technology could be applied not only in the vehicle development phases but also in production and supply chain management. An interview with three major German industry players highlights the challenges of modern supply chains, including their complexity, geographic dispersion, interconnected networks, and diverse regulatory frameworks. These challenges could be addressed by using blockchain as a public digital platform, enabling real-time, transparent connections between multiple supply chain actors. However, the interviews also revealed a significant barrier to adoption: uncertainty surrounding legal regulations, particularly regarding data privacy [44].

In the post-production (after-sales) phase, a blockchain-based continuous monitoring system can assist OEMs in receiving anonymized field data, reducing concerns related to data trust and security [45]. This system periodically transmits information such as diagnostic data codes, performance metrics, and part authenticity from the vehicle to maintenance servers. The data on these servers can be accessed not only by OEM data analysts but also by vehicle owners, enhancing trust in the vehicle’s safety, maintenance quality, and service history [45].

##### Machine Learning and Deep Learning

The application of AI in vehicle security is promising, but still limited due to big memory consumption and processor resources [46]. For these reasons, AI is planned to be used mostly in cloud systems and less in vehicles where embedded processors are used.

Machine learning-based IDSs were proposed for big data analytics in vehicle networks [47,48], while DL techniques could be used for IDS design [49,50].

In a review of the automotive cyber-attacks, V.K. Kukkala proposed an AI-based IDS in IVNs and VANET environments, particularly for Vehicle-to-Vehicle (V2V) and Vehicle-to-Everything (V2X) communication. AI is used as an in-vehicle IDS (GAN-based IDS; GRU-based recurrent autoencoder; LSTM-based encoder–decoder with self-attention; Temporal CNN with neural attention). There are also solutions recommended for actual cybersecurity challenges: data protection and privacy; tamper-proof AI; secure integrated circuit supply chain [20]. In another review, S. Rajapaksha et al. focus on AI-based IDS solutions for CAN communication, highlighting the limitations of these approaches and discussing the security challenges inherent to AI models [46]. Supervised and unsupervised learning (ML or DL) have to be combined with rule-based techniques to cover the complete range of cybersecurity threats. The conclusion of the review is that host-based IDSs are not a practically solution for vehicles (ECU HW limitation), but network-based IDS can be introduced like a separate ECU in vehicles or like a virtual IDS in the cloud (less feasible regarding real-time requirements).

The integration of AI in the automotive industry has led to a need for innovative methods to validate self-learning systems. By combining the benefits of scenario-based testing with metamorphic relations—especially for generating test inputs and creating test oracles—it is possible to mitigate functional and security risks [51].

##### Cybersecurity and Safety Relationship

C.W. Lee, S. et al. applied a cybersafety method (System Theoretic Process Analysis -STPA and STPA-Sec) to analyze safety and security hazards. Later, STPA and CHASSIS (Combined Harm Analysis of Safety and Security for Information Systems) methods were compared. Both methods (STPA and CHASSIS) were applied to the Mobility-as-a-Service (MaaS) and Internet of Vehicles (IoV) cases, focusing on the OTA feature. Their results show that the STPA method (analyzing the system taken as a whole) identified additional hazards and more effective requirements compared to CHASSIS [52].

The fuzz testing method is increasingly being applied in automotive projects for cybersecurity specifications at the system test level. However, in some cases, this method can also be employed for functional and safety-related objectives during early development stages, such as unit testing and integration testing [7].

Possible effects of network-induced delays (resulted from Denial-of-Service cyberattacks) for autonomous vehicles could affect driving safety and comfort. For such a case, Viadero-Monasterio et al. propose a multi-input multi-output method for path tracking control, a method that could attenuate the safety effects for network delays of millisecond order [53].

##### Secure Onboard Communication (SecOC)

The AUTomotive Open System ARchitecture (AUTOSAR) partnership created and established an open and standardized software architecture for automotive ECUs [54]. In 2017, the first specification for secure onboard communication (SecOC) [55] was released, describing a practical approach of how secure in-vehicle communication can be achieved. Instead using a shared key between sender and all receivers, a secret pair consisting of a public key and a secret key is used. In this way, the receiver has the possibility to check the authenticity of sender and also the integrity of the received data. Nowadays, SecOC is implemented by almost all OEMs, and SecOC is used not for all CAN messages, but only for safety-critical and cybersecurity-relevant messages. SecOC implementation requires periodic resynchronization phases, and its implementation across various embedded systems demands significant computational power. Later on, similar specifications were released over AUTOSAR group for secure hardware extensions, crypto stack, secure diagnostics [56] and secure logging, identity and access management, intrusion detection system manager [57], secure updates, and trust platforms.

##### Internet of Things (IoT)

The Internet of Things is increasingly being integrated into the automotive industry, primarily through intelligent sensors [58]. Designed to be lightweight, Message Queue Telemetry Transport (MQTT) was the easiest IoT data communication protocol adopted in the automotive sector.

Previous studies analyzed the impact of adding the Transport Layer Security (TLS) protocol to MQTT communication, with positive results observed in networks with lower busloads [59]. The implementation of TLS and MQTT in automotive networks permit now the secure software update Over-The-Air. Shin et al. [60] propose a novel firmware over-the-air (OTA) update method, MQTree, which combines the MQTT protocol with Merkle tree-based blockchain verification to enhance the efficiency of software updates. The study demonstrates that MQTree performs well against spoofing, man-in-the-middle, and duplicate update attacks. However, to address denial-of-service threats, the addition of a firewall to the system is necessary.

Previous studies regarding IoT analyzed the potential risks to data privacy that can be introduced by this technology, and explored ML and DL solutions to mitigate these risks [61,62]. The conclusions were that current solutions are still in the early stages, and indicated that IoT architecture based on deep neural networks could be adapted for use in cyberattack monitoring systems or IDSs.

##### Automotive Ethernet (AE)

To meet the automotive industry’s requirements for electromagnetic compatibility and immunity, a new Ethernet standard was developed: 100Base-T1, which supports full-duplex operation over a single twisted pair [63]. Previous studies have shown that Automotive Ethernet is gaining traction in vehicle networks due to its high data transfer speeds and enhanced cybersecurity features [64]. These studies also identified security vulnerabilities in Ethernet communication and presented potential countermeasures. De Vincenzi et al. experimentally compared four cybersecurity solutions (SecOC, TLS, Internet Protocol Security, and Media Access Control Security—MACSec) implemented on data link layers of AE [65]. None of the compared solutions excelled in all operation steps. In systems where speed is the most important element, the combination of Advance Encryption Standard and HMAC solutions seems to be the best choice, while in the context where the security is the priority, a combination of Advance Encryption and MACSec provides a good compromise between security and timing.

Two ongoing projects are focused on enhancing the safety properties of automotive Ethernet within the scope of Time-Sensitive Networking (TSN) profiles: IEEE P802.1DG and IEEE P802.DU [66].

##### Data Privacy Challenges

Unlike the IT domain, where security risks mainly target data privacy and confidentiality, the automotive sector—particularly for autonomous vehicles—places a strong emphasis on functional safety. M. Benyahya conducted a review of cybersecurity and data privacy concerns for autonomous vehicles, highlighting the need for greater involvement from automotive stakeholders [67]. Technical mitigation solutions, such as data anonymization and encryption, should be incorporated during case study and implementation phases of vehicle development. Despite ongoing advancements in Zero Knowledge Theory as a method for encrypting private data, there is still a gap in legislative measures, particularly the absence of a trusted authority for secure key exchange between different users [68].

A privacy manager was developed as a technical solution for data privacy, aiding applications by filtering specific data in real-time to support the implementation of privacy functions [69]. This approach was validated using two scenarios: platooning and silence testing, with GPS position data from the vehicle serving as the test input.

Mobility as a Service (MaaS) applications impose strict requirements for data privacy. Kong et al. [70] explore a privacy-preserving solution for driver monitoring using blockchain technology, specifically encryption with public keys. The practical aspect of the study demonstrates the feasibility of this approach, though it highlights that the most time-consuming phase is data transfer between the vehicle and the cloud. Additionally, a limitation of the method is its scalability, particularly when handling real-time data acquisition from multiple vehicles.

##### Cloud Cybersecurity Solutions

A review of the cybersecurity requirements for cloud-supported connected vehicle (CV) applications identified six key categories: confidentiality, integrity, privacy, authentication, accountability, and availability. Several cybersecurity challenges were highlighted, including the authentication of high mobility nodes, trustable V2V communication, vehicle location validation, securing in-vehicle network (IVN) communication, and data privacy in the cloud. Future research directions include exploring Infrastructure as Code, integrating blockchain with AI, leveraging quantum computing for machine learning, utilizing 5G for secure and faster communication, heterogeneous wireless networking, and network function virtualization [43].

### 3.2. CS Standards and Regulations

ISO 26262 was the first standard developed for managing functional safety in automotive applications [71]. Its primary goal is to establish a uniform approach for OEMs to address safety considerations.

The increasing complexity of high connectivity interfaces, shared services, and advanced autonomous vehicle features has necessitated a shift in the development of the E/E systems, emphasizing the need for enhanced cybersecurity measures. In 2018, a revised version of ISO 26262 [11] was released, introducing significant updates such as improved management of safety anomalies, more detailed objectives, references to cybersecurity, and additional requirements for trucks, buses, and trailers. This was preceded by the introduction of SAE J3061 in January 2016, the first global cybersecurity standard for the automotive industry [8].

Recently, the automotive sector has placed a stronger emphasis on cybersecurity while continuing to address safety functions. New standards, norms, and guidelines are being developed to establish unified requirements for both cybersecurity and functional safety and to provide practical tools for their implementation. Table 1 provides a summary of the most relevant standards, norms, and guidelines in this field.

Previous studies explored the integration of safety and cybersecurity for risk management and system/software validation [83,84,85]. ISO 21434 provides a structured framework for cybersecurity, detailing a uniform development process, specific requirements and specifications, and a standardized language for communicating and managing cybersecurity risks among stakeholders [86,87].

A cybersecurity development lifecycle model, incorporating both ISO 21434 and Cybersecurity ASPICE requirements, has been proposed and implemented in automotive pilot projects, such as those involving electric power steering systems [88]. This model emphasizes threat modeling and vulnerability analysis as fundamental components.

The introduction of ISO 24089 aims to transform the management of software updates within the automotive sector [82]. Alongside ISO 24089, UN Regulation No. R156 [78] has established a more standardized approach to software update requirements through the Software Update Management System (SUMS).

Schober et al. [89] reviewed the current state of automotive cybersecurity regulations and standards, highlighting the connections and interdependencies among them.

### 3.3. CS Monitoring and Vulnerability Management

#### 3.3.1. Intrusion Detection Systems

Intrusion Detection Systems (IDSs) have recently been incorporated into automotive architectures alongside cybersecurity risk mitigation strategies. While IDS is not explicitly mentioned in ISO 21434 or R155 regulations, various IDS approaches are frequently analyzed and proposed for detecting attacks, logging incidents, and mitigating threats. Most IDS solutions are concentrated on CAN and Ethernet communication channels [90,91].

IDS functionalities can be implemented either as a separate or dedicated ECU, within the vehicle network (network-based IDS). This could involve a cloud-based software solution that receives data from the vehicle’s telematics ECU [92] (cloud network-based IDS), or a software module embedded in one or more high-performance ECUs within the vehicle (host-based IDS) [93]. Currently, OEMs are more inclined to deploy dedicated ECU IDS versions.

In terms of attack detection methods, IDS systems are categorized into signature-based or anomaly-based systems [94]. Signature-based IDS utilize a database of known attack signatures and monitor permissible data exchanges between ECUs. This database requires regular updates, as does the information regarding allowed data exchanges following any ECU software updates. Anomaly-based IDS, on the other hand, monitor changes in physical properties (e.g., voltage, current, busload) of in-vehicle network communications [95,96], or detect anomalies in functional signal values (e.g., GPS signal jumps indicative of spoofing attacks) [97].

Researchers are also focusing on in-vehicle network IDS as a robust defense mechanism against automotive attacks [98]. IDS solutions are being explored for Vehicular Ad-hoc Network (VANET) systems [99], including flow-based IDS that are sensitive to timing and frequency changes of messages, and payload-based IDS that detect modifications to message content [100]. Furthermore, algorithms are being developed to generate real-time model parameters for specific CAN buses, enabling specification-based IDS using anomaly-based supervised learning with real-time models (SAIDuCANT) [101].

Mansourian et al. [102] propose an IDS solution that integrates flow-based and payload-based IDS through three modules: a time-based prediction network, a payload-based prediction error processor, and a Gaussian Naïve Bayes classifier to determine the presence of active attacks.

The data-driven, payload-based method for identifying falsified vehicle functionalities, such as compromised connected vehicle trajectories, has been extensively researched. This method can also be applied to detect safety-critical events [103].

For CAVs requiring communication with external systems like cloud services or road infrastructure (e.g., platooning), developing external or internal IDS supported by firewalls could be beneficial [104,105]. Park et al. [104] demonstrate a method for detecting adware and malware in Android OS-based ECUs. Additionally, a study on the Service-Oriented Architecture (SOA) paradigm [105] presents a combination of firewall, IDS, and Identity and Access Management (IAM) as countermeasures to protect autonomous vehicles.

With the growing intelligence of vehicle sensors, which now perform not just analog-to-digital conversions but also digital signal processing, and the integration of IoT technologies, sensors interact with ECUs using standard communication protocols. Research is underway to identify potential threats introduced by smart sensors, their associated security risks, and methods for cybersecurity monitoring and attack detection at the sensing layer [58].

#### 3.3.2. Security Operation Center

New regulations are requesting OEMs to establish dedicated vehicle security teams responsible for monitoring all sold vehicles and addressing new vulnerabilities discovered in their software. The study referenced in [106] identifies commonalities and differences between Security Operation Centers (SOC) in IT and Vehicle Security Operation Centers (VSOC). It highlights that methods, procedures, and technical solutions from IT SOCs cannot typically be applied directly to VSOCs due to their unique requirements.

In other studies, Fenzl et al. examined various methods for reporting incidents to both internal and external SOCs, focusing on how collaborative security patterns can be developed [107], while Barletta et al. present a methodology that integrates quantum optimization with intrusion detection systems and the National Vulnerability Database. This approach allows intrusion systems to learn from incidents reported and added to the vulnerability database [108].

#### 3.3.3. Threat Analysis and Risk Assessment (TARA)

Wang et al. [109] propose a systematic risk assessment framework that includes a specific risk assessment process and systematic methods. This framework has a similar structure with TARA process mentioned in ISO 21434 [10], and it is based on three standard blocks: risk identification; risk analysis; and risk assessment, that are applicable across all phases of the vehicle lifecycle.

Zhang et al. conducted a case study focusing on threat analysis, test item determination, and vulnerability scoring, which are essential components of a comprehensive TARA procedure [110].

Dobaj et al. carried out a case study on risk-driven system design and the development of cybersecurity requirements [111]. This study complements the standard and offers engineers a practical guide for developing secure systems. Prior to initiating TARA, preliminary steps must be taken, including identifying system assets and associated risks through structured approaches.

A scenario-based TARA approach was also introduced [112], which was applied to Over-the-Air updates for CAVs to derive cybersecurity goals, which will then be translated into cybersecurity requirements.

Neither systematic risk assessment (typically used in model-based development projects) nor scenario-based risk assessment provides a complete solution for all automotive applications [31]). Depending on the safety criticality and complexity of the application, one of these methods might be chosen, or in some cases, a combined approach may be used, which could increase the costs of product development.

The Trusted Information Security Assessment Exchange (TISAX) certification, which is adapted from ISO 27001 [113], serves as a European automotive industry standard for “information security assessment” (ISA). It focuses on key aspects of information security, such as data protection and third-party connections [114]. The term TISAX did not appear in our bibliographic research because it is associated with “security” rather than “cybersecurity” within the automotive context. The implementation methods and benefits of TISAX were analyzed in [115]. A new version of TISAX is expected to be released in 2024.

#### 3.3.4. Cybersecurity and Platooning Functions

Cybersecurity is crucial for platooning functions and connected vehicles [116]. Various studies have examined the impact of cybersecurity attacks on vehicle platoons [17,117,118,119]. Additionally, several cybersecurity solutions have been proposed for vehicle platoons, including an anomaly detection system [120] and a method for detecting false data injection attacks [121].

#### 3.3.5. Secure Communication

Secure communication is an effective approach to enhancing the cybersecurity of vehicle systems. Since 2014, AUTOSAR has provided specifications for the Secure OnBoard Communication (SecOC) function [55], which can now be seamlessly integrated into ECU software by any supplier using AUTOSAR stacks and libraries.

To further bolster the cybersecurity of IVNs, various secure communication protocols have been proposed and validated, including TOUCAN and IDH-CAN [18,122,123,124,125,126]. TOUCAN (proTocol tO secUre Controller Area Network) was designed to secure CAN communication [122]. An improved version of TOUCAN, called CINNAMON (Confidential, INtegral aNd Authentic on board co-MunicatiON), was proposed to be compatible with AUTOSAR and addresses confidentiality issues that SecOC does not [127,128].

Identification Hopping CAN (IDH-CAN) is another hardware and software solution that ensures both communication security and real-time application constraints. It introduces a hardware firewall between the physical and data link layers. However, implementing IDH-CAN in existing vehicle architectures can be challenging, as it requires simultaneous updates to all hardware CAN drivers [124].

Palaniswamy et al. developed a new secure CAN protocol suite that includes a session key update protocol (NSKUP) to prevent key reuse [18,125]. While NSKUP requires significant computational resources and introduces time delays, Groza et al. introduced a lightweight broadcast authentication (LiBrA-CAN) protocol as a more resource-efficient alternative [129].

Additionally, the security of wireless communication between On-Board Units (OBUs) and Road Side Units (RSUs) has been examined, with recommendations for implementing lightweight cryptographic techniques for CAVs [126]. For vehicles in motion, rapid initialization of wireless communication between OBUs and RSUs is essential, as standard Wi-Fi authentication methods are too slow for Vehicle-to-Infrastructure (V2I) communication.

### 3.4. CS Testing and Validation

To enhance cyber defense activities, it is essential to advance validation and verification processes. This includes improving testbenches, CAN simulators, and Hardware-in-the-Loop systems, developing new types of test cases, analyzing in-vehicle firewall properties, and exploring innovative cybersecurity solutions [130,131].

In support of the secure-by-design principle, methods adopted from other industries (black-box fuzz testing) has been developed to construct effective security tests. A case study with a CAN-fuzzer demonstrate the effectiveness of fuzz testing [132], while a security analysis and a reverse engineering case study applied for a instrument cluster ECU shown a way of using fuzz testing in later phases of product development [133]. Future research should focus on optimizing how useful metrics are gathered from fuzz testing.

Digital Twin-Based Security Testing is a new method proposed, which involves creating and executing cybersecurity test cases automatically in a black-box setting [134]. This approach demonstrates the feasibility of test automation by transferring attacks from a model (one system) to a SUT, through generalization using a domain-specific language. A case study using digital twin-based security method could not be identified in the literature.

In the past decade, model-based testing methods (whitebox testing) have become common in the automotive industry. These methods can also be applied to automotive security testing as a partial solution for detecting vulnerabilities [29,30,31].

Cui et al. [135] developed a simulation platform to s evaluate the performance of specific autonomous vehicle functions and conduct safety impact analyses under various cybersecurity attack scenarios.

Marksteiner et al. [136] conducted a case study that incorporates not only functional testing methods but also interface testing, static code analysis, penetration testing, vulnerability scanning, and fuzz testing. The proposed structured testing process is adaptable, allowing test engineers to use their preferred toolsets.

Standardizing automotive penetration testing from a black-box perspective is challenging. Zhang et al. developed a case study for a penetration testing framework called ICVTest, which guides inexperienced testers step-by-step through test case generation and execution [137].

Risk management during the development phase is handled similarly for safety and cybersecurity, using HARA (Hazard Analysis and Risk Assessment) for safety and TARA for cybersecurity [10]. Some studies suggest that integrating safety and security risk management, including shared processes and documentation, could provide a better overview of product risks and potentially reduce costs [52,84,112].

Model-based testing is not only applicable to functional safety requirements but can also partially support cybersecurity testing [28], helping to identify security issues earlier in the development process (as penetration testing typically occurs later).

Additionally, studies [138,139] propose sensor solutions for automobiles that enhance both safety (real-time and recovery requirements) and cybersecurity (attack detection) performance simultaneously.

To improve effectiveness and identify bugs earlier in the development process, Oka et al. suggest moving fuzz testing from the System and Acceptance Test phases to the Unit Test phase [140] This change would also facilitate easier automation and execution of fuzz tests.

## 4. Conclusions and Future Research

This paper provides a comprehensive review of the current state and challenges of [138] cybersecurity in the automotive industry. The bibliometric network analysis was based on the Web of Science Core Collection database and highlighted four clusters of cybersecurity-related themes: frameworks and technologies, standards and regulations, monitoring and vulnerability management, and testing and validation.

**Frameworks and Technologies**. Current automotive development frameworks are beginning to incorporate comprehensive cybersecurity processes required by new standards. There is a need to optimize these processes and improve their integration throughout the development cycle [15].

Recent studies recommend adopting a system-of-systems approach, which includes not only the vehicle and its components but also the cloud where vehicle data are stored, interactions with other vehicles, road infrastructure, and charging stations for electric vehicles [18].

The application of AI in vehicle security is promising, and the proposed solutions should undergo extensive testing [21,141]. Although AI concepts and algorithms were proposed as solutions for security challenges during the last years, their practical implementation has been limited.

The integration of AI in the automotive industry has led to a need for innovative methods to validate self-learning systems. By integrating the advantages of scenario-based testing with metamorphic relations—particularly for generating test inputs and creating test oracles—it becomes possible to mitigate both functional and security risks [51].

**Standards and Regulations**. The R155 and ISO 21434 regulations mark a significant milestone in establishing clear requirements for the scope, performance, and auditing of cybersecurity, covering the entire product lifecycle [136]. A second release of ISO/SAE 21434 is currently in development, aiming to address gaps in the existing version [125]. Simultaneously, researchers and practitioners must develop specific technologies and methods, such as cybersecurity testing, to help OEMs and suppliers meet the standards’ requirements.

Enhancements are also needed in the relationships between OEMs, suppliers, and third-party providers, as recent cybersecurity incidents have highlighted vulnerabilities originating from ECU supplier software [130]. The first edition of ISO/SAE 21434 provides a good framework for dividing responsibilities in security incidents, but agreements between OEMs and suppliers, or between suppliers and third parties, lack standardization.

**Monitoring and vulnerability management**. While many IDS solutions have been proposed, OEMs have been reluctant to adopt them, primarily due to implementation costs and real-time performance issues [142]. There is a need to extend the validation of proposed IDSs across different vehicle configurations and architectures. Many existing IDS solutions are designed for CAN networks, and significant changes may be required for CAN-FD. Future research should also focus on recovery strategies for IVNs, after a security breach is detected [101].

The theoretical frameworks for cybersecurity risk assessment (TARA) need also refinement to develop practical models and methods. Future research should aim to enhance risk assessment indicators [107] and create a generic security-driven development lifecycle model [111].

CAVs face multiple conflicts, such as balancing safety with security and cost with usability. These conflicts should be identified in the early design phases to mitigate their impact [112]. The effects of cyberattacks in mixed traffic scenarios involving CAVs and human-driven vehicles need further analysis [52].

**Previous studies** focused on improving communication security while minimizing the impact on real-time automotive processes [128]. Lightweight secure communication can be a solution to this challenge [129]. Additionally, researchers are exploring hardware solutions to enhance the temporal performance of secure communication.

**Testing and Validation**. The addition of new cybersecurity features has increased the complexity of testing, particularly in simulation tools and test execution [134]. Some proposed testing solutions have only been validated in small-scale setups with limited ECUs [131]. A significant research gap remains in the development of effective metrics for fuzz testing [132].

Cybersecurity testing often emphasizes automated test case generation to meet requirements. However, penetration testing remains a manual method, and adapting fuzz testing from IT to the automotive sector has potential for improvement, especially in testing efficiency.

Despite significant progress, there remains a limited number of studies addressing cybersecurity throughout the entire automotive lifecycle. Future research is essential given the growing complexity of cyber-physical systems, the expanding attack surfaces, the increasing demand for data protection, the integration of AI in autonomous vehicles, and the critical importance of supply chain security and regulatory compliance.

## Figures and Tables

**Figure 1 sensors-24-06139-f001:**
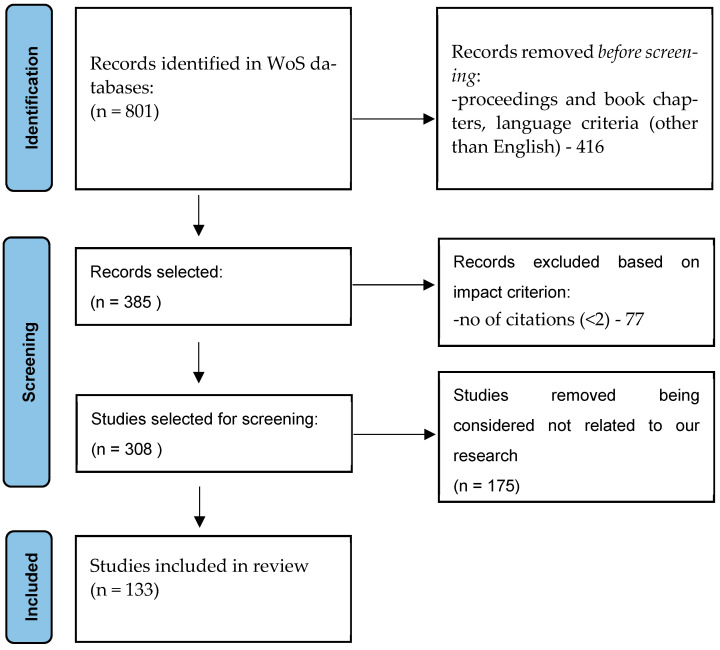
PRISMA flow diagram outlining the study selection process.

**Figure 2 sensors-24-06139-f002:**
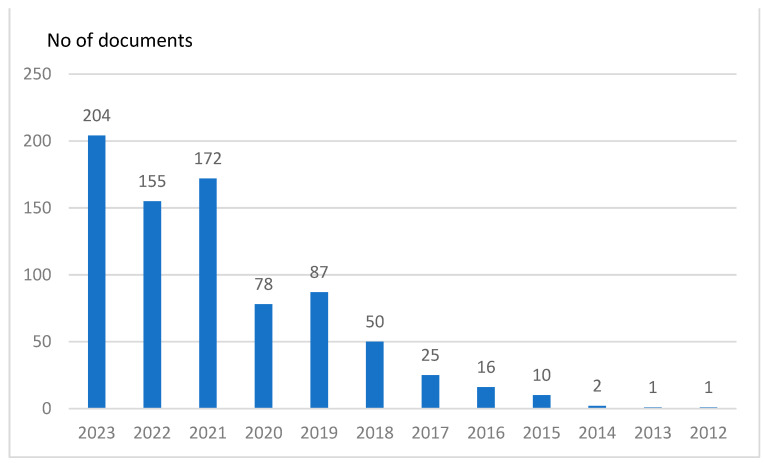
Number of publications per year from the WoS database (based on search: “cybersecurity” + “vehicle” or “cybersecurity” + “automotive”).

**Figure 3 sensors-24-06139-f003:**
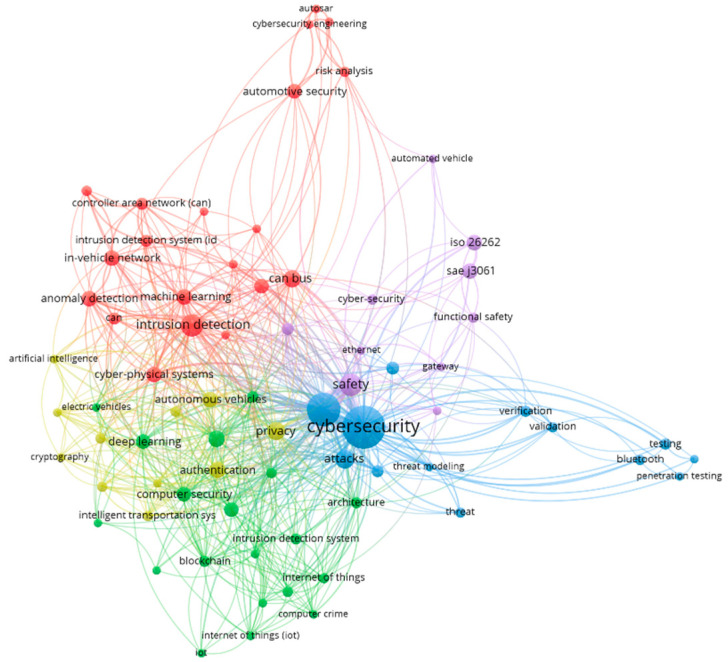
VOSViewer clusters for automotive cybersecurity paper keywords.

**Table 1 sensors-24-06139-t001:** Standards, norms and guideline for functional safety and cybersecurity.

Standard/Norm/Guideline	Name
ISO 26262:2011 [71]	Road vehicles—Functional safety
AUTOSAR (2014) [72]	AUTOSAR safety solutions
SAE J3061:2016 [8]	Cybersecurity Guidebook for Cyber-Physical Vehicle Systems
ASPICE (2017) [73]	Automotive SPICE Process Reference and Assessment Model
ISO 26262:2018 [11]	Road vehicles—Functional safety
TR-68 (2019) [74]	Technical reference. Autonomous vehicles (Singapore Standards Council)
ISO/TR 4804:2020 [75]	Road vehicles—Safety and cybersecurity for automated driving systems—Design, verification and validation
SAE J3061:2021 [76]	Cybersecurity Guidebook for Cyber-Physical Vehicle Systems
ISO/SAE 21434:2021 [10]	Road vehicles—Cybersecurity engineering
ASPICE for Cybersecurity (2021) [77]	Automotive SPICE for Cybersecurity
UN R155 (2021) [9]	Cyber security and cyber security management system
UN R156 (2021) [78]	Software update and software update management system
UN R157 (2021) [79]	Automated Lane Keeping Systems
ISO 21448:2022 [13]	Road vehicles—Safety of the intended functionality
AUTOSAR (2022) [80]	AUTOSAR cybersecurity solutions
ISO/PAS 5112:202) [81]	Road vehicles—Guidelines for auditing cybersecurity engineering
ISO/SAE 24089:2023 [82]	Road vehicles—Software update engineering

## Data Availability

Not applicable.

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
