# Peer review of "Automotive Cybersecurity: A Survey on Frameworks, Standards, and Testing and Monitoring Technologies"

_sensors, 2024, doi:10.3390/s24186139_

Round 1

Reviewer 1 Report

Comments and Suggestions for Authors

This article is a comprehensive and detailed review of automotive network security research, which not only analyzes the current research status in this field, but also proposes practical suggestions and future research directions for the identified problems. The article uses VOSviewer tool for literature analysis, dividing relevant research into four main categories, including framework and technology, standards and regulations, monitoring and vulnerability management, and testing and validation. This structural arrangement helps readers systematically understand the different aspects of automotive network security. At the same time, the article emphasizes the challenges faced in finding practical solutions to these problems, pointing out the significant challenges that software engineers face in diverse communication channels, software integration, complex testing, and other aspects. But there are still some issues that need to be discussed in detail in my review, so that readers can better understand the specific and difficult problems currently faced in this field?

1. Although the article covers multiple aspects of automotive network security, it can further expand on the content of IoT security threats (communication transmission security, protocol security, content security, data reliability security, etc.), as well as the current status and development of security research in these areas internationally.

2、The article proposes some risk assessment frameworks and testing methods, but lacks specific case studies on the application of these methods. It is recommended to add one or more case studies to demonstrate the practical effectiveness and differences of these methods, enhancing readers' operability and practicality in this area of application and security analysis.

3、Given the increasingly widespread application of artificial intelligence and machine learning in automotive network security, this article has not conducted a comprehensive review and analysis in this area. It is necessary to increase the analysis of the security challenges and solutions that these technologies may bring, as well as provide reasonable suggestions for solving some prominent problems?

4、The article proposes many suggestions on automotive network security, but lacks detailed implementation steps and operational guidelines. To improve practicality, it is recommended to provide more specific security measures and execution strategies? Can readers or researchers be inspired by it, especially by analyzing some specific implementation methods and cases that have already been carried out internationally? Compare the advantages and disadvantages of these methods?

5、As cars increasingly collect and process personal data, privacy protection and ethical use of data have become particularly important. The article did not explore these issues and how to respect user privacy while ensuring security. Please add relevant content and analysis suggestions?

Comments on the Quality of English Language

1、The frequent use of professional terminology in the article may cause comprehension barriers for non professional readers. Please provide appropriate explanations (by adding corresponding tables or simplifying these terms).

2、Figure 2 in the paper is not clear and cannot be seen clearly.

3、Most of the literature reviews in the paper are explained in text, which is not very reasonable and has poor readability and comparability.

Author Response

Dear reviewer,

Thank you very much for taking the time to review our manuscript. Please find the detailed responses below and the corresponding revisions/corrections highlighted/in track changes in the re-submitted files.

In response to reviewer’s 2 suggestion (...the authors might consider narrowing their focus …), we decided to reformulate the title as:

Cybersecurity in automotive: A survey on frameworks, standards, and testing and monitoring technologies

General comments

This article is a comprehensive and detailed review of automotive network security research, which not only analyzes the current research status in this field, but also proposes practical suggestions and future research directions for the identified problems. The article uses VOSviewer tool for literature analysis, dividing relevant research into four main categories, including framework and technology, standards and regulations, monitoring and vulnerability management, and testing and validation. This structural arrangement helps readers systematically understand the different aspects of automotive network security. At the same time, the article emphasizes the challenges faced in finding practical solutions to these problems, pointing out the significant challenges that software engineers face in diverse communication channels, software integration, complex testing, and other aspects. But there are still some issues that need to be discussed in detail in my review, so that readers can better understand the specific and difficult problems currently faced in this field?

Response. Thank you for your comments and suggestions, they helped us to improve the manuscript.

C1. Although the article covers multiple aspects of automotive network security, it can further expand on the content of IoT security threats (communication transmission security, protocol security, content security, data reliability security, etc.), as well as the current status and development of security research in these areas internationally.

R1. Thank you for this suggestion. We agree with this comment. Therefore, we have expanded the discussions on IoT, with new paragraphs (starting with line 321)

C2. The article proposes some risk assessment frameworks and testing methods but lacks specific case studies on the application of these methods. It is recommended to add one or more case studies to demonstrate the practical effectiveness and differences of these methods, enhancing readers' operability and practicality in this area of application and security analysis.

R2. Thank you for pointing this out. We agree with this comment. More case studies were integrated into the manuscript (lines 533, 668, 677, 752, 767)

C3. Given the increasingly widespread application of artificial intelligence and machine learning in automotive network security, this article has not conducted a comprehensive review and analysis in this area. It is necessary to increase the analysis of the security challenges and solutions that these technologies may bring, as well as provide reasonable suggestions for solving some prominent problems?

R3. Thank you for pointing out this comment. As other studies (please see Rajapaksha, S. 2023) performed detailed analysis on AI in automotive, we choose not to go into detail in this direction. However, based on your suggestion we mentioned this study in our article (line 295).

C4. The article proposes many suggestions on automotive network security, but lacks detailed implementation steps and operational guidelines. To improve practicality, it is recommended to provide more specific security measures and execution strategies? Can readers or researchers be inspired by it, especially by analyzing some specific implementation methods and cases that have already been carried out internationally? Compare the advantages and disadvantages of these methods?

R4. Thank you for pointing this out. Unfortunately, the literature does not provide such detailed implementation steps and operational guidelines. We hope that the case studies (mentioned before) together with the examples and also the other changes we introduced in the paper, provide a more realistic view of different practical approaches and the of advantages and disadvantages of different methods

C5. As cars increasingly collect and process personal data, privacy protection and ethical use of data have become particularly important. The article did not explore these issues and how to respect user privacy while ensuring security. Please add relevant content and analysis suggestions

R5. Thank you for pointing this out. We added more content on data privacy, especially in the Conclusion section (starting with line 902)

Other comments

C.i. The frequent use of professional terminology in the article may cause comprehension barriers for non professional readers. Please provide appropriate explanations (by adding corresponding tables or simplifying these terms).

R.i. Thank you for pointing this out. We tried to simplify the terms used in the paper, and reformulated some sentences, concepts

C.ii. Figure 2 in the paper is not clear and cannot be seen clearly.

R.ii. Thank you for pointing out this suggestion. We replaced Figure 2 with a better-quality one

C.iii. Most of the literature reviews in the paper are explained in text, which is not very reasonable and has poor readability and comparability.

Riii. Thank you for pointing this out. We reformulated many sentences for better readability and clarity

Reviewer 2 Report

Comments and Suggestions for Authors

Automotive cybersecurity is a dynamic and rapidly evolving research field. This study provides a review of automotive cybersecurity, particularly focusing on Cyber Security Management Systems. It organizes research into frameworks, standards, monitoring, and testing, highlighting key findings and proposing future research directions. However, there are a few areas that could be improved:

  1. Automotive cybersecurity is a broad field, so the authors might consider narrowing their focus to a specific area for a deeper investigation. Here are some examples: https://www.sciencedirect.com/science/article/pii/S0167404824003237; https://ieeexplore.ieee.org/abstract/document/10339926.

  2. As a survey paper, it should do more than just list existing research. It’s important to classify, compare, and analyze the findings to give readers valuable references for their research.

  3. The authors should consider including papers published in 2023 and 2024 to ensure the study reflects the most recent developments.

  4. The proposed future directions could be more detailed and actionable to better guide further research.

Comments on the Quality of English Language

Please give a detailed description of the papers you surveyed.

Author Response

Dear reviewer,

Thank you very much for taking the time to review our manuscript. Please find the detailed responses below and the corresponding revisions/corrections highlighted/in track changes in the re-submitted files.

Automotive cybersecurity is a dynamic and rapidly evolving research field. This study provides a review of automotive cybersecurity, particularly focusing on Cyber Security Management Systems. It organizes research into frameworks, standards, monitoring, and testing, highlighting key findings and proposing future research directions. However, there are a few areas that could be improved

Thank you for your comments and suggestions, they really helped us to improve our manuscript

C1. Automotive cybersecurity is a broad field, so the authors might consider narrowing their focus to a specific area for a deeper investigation. Here are some examples: https://www.sciencedirect.com/science/article/pii/S0167404824003237;

https://ieeexplore.ieee.org/abstract/document/10339926.

R1. Thank you for pointing this out. We agree with this comment, that why we reformulated the article title as Cybersecurity in automotive: A survey on frameworks, standards, and testing and monitoring technologies

C2. As a survey paper, it should do more than just list existing research. It’s important to classify, compare, and analyze the findings to give readers valuable references for their research.

R2. Thank you for your suggestion. We actually have (in an Excel table) such a detailed description. But instead of including it in a separate Annex we decided to integrate it into the manuscript, for better readability

C3. The authors should consider including papers published in 2023 and 2024 to ensure the study reflects the most recent developments.

R3. Thank you for pointing this out. We agree with this comment. Six more references (from the last years) were added and cited in the paper.

C4. The proposed future directions could be more detailed and actionable to better guide further research.

R4. Thank you for pointing this out. We agree with this comment.  We added more content in the Conclusions and future research perspectives section

Round 2

Reviewer 1 Report

Comments and Suggestions for Authors

The paper has undergone some minor revisions based on the comments of the reviewers, but the quality of the revisions is still unsatisfactory.

1) In response to the reviewer's comments, many authors only added a sentence or cited a reference without conducting substantive analysis and discussing its advantages and disadvantages? Therefore, it is recommended to make reasonable revisions based on the previous opinions and supplement the relevant content completely.

2) Is the response to the revised manuscript unclear, making it difficult for reviewers to find the corresponding position in the revised manuscript?

Comments on the Quality of English Language

The modifications are basically in place and meet the publishing requirements

Author Response

Dear reviewer,

Thank you for your suggestions, they really helped us to improve our paper. Please find attached an updated version of the manuscript that hopefully fulfills all of your requirements. To provide a more "readable" document, we removed the comments from the first round of review.

General comment. The paper has undergone some minor revisions based on the comments of the reviewers, but the quality of the revisions is still unsatisfactory.

R. There are substantial revisions in our paper now, after the second round of review. These changes are concerning especially the Chapter no 3. Literature review, where new sub-sections were introduced, and many paragraphs were added or modified. Changes were operated also in other chapters of the article. Compared to the first version of the manuscript, 15 more references were introduced and discussed, to respond to reviewers' suggestions.

We completely restructured and rewritten the Conclusions and Future Research section.

1) In response to the reviewer's comments, many authors only added a sentence or cited a reference without conducting substantive analysis and discussing its advantages and disadvantages? Therefore, it is recommended to make reasonable revisions based on the previous opinions and supplement the relevant content completely.

R.1. We hope that this version fulfills all the requirements

2) Is the response to the revised manuscript unclear, making it difficult for reviewers to find the corresponding position in the revised manuscript?

All these changes were made using Track changes facilities, as the journal editorial team suggested. 

Reviewer 2 Report

Comments and Suggestions for Authors

It’s challenging to identify the improvements made in your revisions. I suggest providing a clean version of the manuscript (without comments) alongside the commented version for easier comparison.

Additionally, a survey should offer more than just a collection of published works; it should provide meaningful insights and implications to guide readers in their own research. Please enhance the discussion of your references and clarify the key takeaways you want to convey.

The section on Future Research Directions reads as if it was generated by AI software. I recommend thoroughly revising it to ensure originality and depth.

Author Response

Dear reviewer,

Thank you for your comments and suggestions. Please find attached an updated version of the manuscript that hopefully fulfills all your requirements. In order to provide a more "readable" document, we removed the comments from the first round of review.

General comment. It’s challenging to identify the improvements made in your revisions. I suggest providing a clean version of the manuscript (without comments) alongside the commented version for easier comparison.

R.1. You are right, the article is difficult to read with so many changes. All these changes were made using Track changes facilities, as the journal editorial team suggested. And the journal also asks to leave the changes visible. A clean copy can be obtained by hiding all these changes. We provided also a pdf version of the manuscript cleaned, with comments removed

C1. Additionally, a survey should offer more than just a collection of published works; it should provide meaningful insights and implications to guide readers in their own research. Please enhance the discussion of your references and clarify the key takeaways you want to convey.

R1. You are perfectly right. Compared to the first version of the manuscript, 15 more references were introduced and discussed, to respond to reviewers' suggestions. Chapter 3 was substantially modified - new subtitles were introduced, new paragraphs were added and corrections were made. And we insisted on discussing the references, as you suggested

C2. The section on Future Research Directions reads as if it was generated by AI software. I recommend thoroughly revising it to ensure originality and depth.

R2. It is our work. We can prove, if necessary, by sending the previous 4 - 5 versions of the document.

We agree that this section's style and content are not very good. That is why we completely restructured and rewritten the Conclusions and Future Research section.